# Language Models for Multimessenger Astronomy

**Vladimir Sotnikov** [1,*] and **Anastasiia Chaikova** [2]

1   JetBrains and Astroparticle Physics Lab, JetBrains Research, Paphos 8015, Cyprus
2   School of Computer Science & Engineering, Constructor University, 28759 Bremen, Germany
*   Correspondence: vladimir.sotnikov@jetbrains.com

**Abstract:** With the increasing reliance of astronomy on multi-instrument and multi-messenger observations for detecting transient phenomena, communication among astronomers has become more critical. Apart from automatic prompt follow-up observations, short reports, e.g., GCN circulars and ATels, provide essential human-written interpretations and discussions of observations. These reports lack a defined format, unlike machine-readable messages, making it challenging to associate phenomena with specific objects or coordinates in the sky. This paper examines the use of large language models (LLMs)—machine learning models with billions of trainable parameters or more that are trained on text—such as InstructGPT-3 and open-source Flan-T5-XXL for extracting information from astronomical reports. The study investigates the zero-shot and few-shot learning capabilities of LLMs and demonstrates various techniques to improve the accuracy of predictions. The study shows the importance of careful prompt engineering while working with LLMs, as demonstrated through edge case examples. The study's findings have significant implications for the development of data-driven applications for astrophysical text analysis.

**Keywords:** neural network; natural language processing; large language model; astronomical report; multi-messenger astronomy

## 1. Introduction

As multi-instrument and multi-messenger observations become increasingly important in astronomy, effective communication among astronomers is crucial. In addition to messages for prompt follow-up observations, human-written reports such as GCN circulars and ATels provide in-depth analysis and interpretation of observations. However, the highly technical and unstructured language of these reports makes them difficult to process automatically.

The Astronomer's Telegram (ATel) [1] is a freely available online database of astronomical research and discoveries of transient sources written in natural language. The data contained in an ATel entry usually includes basic information about the object, such as its name, coordinates, and event type, as well as more detailed information, such as its frequency band, energy, spectral and temporal evolution, and the results of follow-up observations. The data can also include images and tabular data, as well as references to related papers and other ATel telegrams.

GCN Circulars is a dataset of astronomical circulars released by the Gamma-ray Coordinates Network (GCN) [2]. The GCN system distributes two types of messages: notices, which contain information about the location of GRBs (gamma-ray bursts) and other transients obtained by various spacecraft, and circulars, which contain information about follow-up observations made by ground-based and space-based optical, radio, X-ray, and TeV observers.

ATels are typically released within hours or days of an observation and provide a quick overview of the discovery and initial analysis. On the other hand, GCN circulars are more comprehensive reports released after a more thorough analysis of the observations. Both types of reports are crucial sources of information for astronomers and astrophysicists

studying transient phenomena such as supernovae, gamma-ray bursts, and others. An example report is shown in Figure 1.

However, these reports' unstructured and often highly technical language can make them difficult to process automatically. This is a natural language processing task that requires advanced techniques, such as the use of large language models (LLMs), to extract relevant information. By using machine learning to analyze and interpret these reports, researchers can more efficiently process and understand the vast amount of data being generated in the field of astrophysics. The successful application of LLMs to analyzing ATels and GCN circulars could facilitate the rapid and accurate interpretation of transient astrophysical phenomena.

```
Subjects:  Gamma Ray, Request for Observations, AGN, Blazar, Transient.
Description:  Referred to by ATel #:  8706, 8718, 8783, 8789 On Jan 14,
2016, the Large Area Telescope (LAT), one of the two instruments on the
Fermi Gamma-ray Space Telescope, observed strong gamma-ray emission
from a new source.  The best-fit location of this gamma-ray source
(RA=8.91 deg, Dec=61.52 deg, J2000.0) has a 95% containment radius
of 0.08 deg (errors are statistical only).  This source is not in
any published LAT atalog and in the past has not been detected by
AGILE or EGRET. The closest candidate counterpart is the radio source
87GB 003232.7+611352 , with coordinates RA=8.8542 deg, Dec=61.5083 deg
(J2000.0; Petrov et al.  2006, AJ, 131 1872), at an angular distance
of 0.03 deg.  Preliminary analysis indicates that on Jan 14, 2016, the
daily-averaged flux (E>100MeV) was (5.7+/-1.5)10^-7 photons cm^-2
s^-1, with a photon index of 1.8+/-0.2 (errors are statistical only).
Because Fermi operates in an all-sky scanning mode, regular gamma-ray
monitoring of this source will continue.  We encourage further
multifrequency observations of this source.  For this source the Fermi
LAT contact person is Giovanna Pivato (giovanna.pivato@pi.infn.it).
The Fermi LAT is a pair conversion telescope designed to cover the
energy band from 20 MeV to greater than 300 GeV. It is the product of
an international collaboration between NASA and DOE in the U.S. and
many scientific institutions across France, Italy, Japan and Sweden.
```

**Figure 1.** An example message from ATel. Named entities are marked with color: ■ object name; ■ the type of the object or physical phenomena; ■ event type. Taken from Ref. [1].

Our research aims to develop a technique for extracting structured information from human-written astrophysical reports and make it easily accessible to the astrophysical community. From these astronomical observations, we aim to extract the following named entities: event ID or object name, observed event (e.g., gamma-ray burst), observed object, or physical phenomena (e.g., BL Lac object, supernova explosion). However, an observation does not necessarily contain all these entities, and/or it may contain mentions of other objects or events, information that is not needed in the output.

To tackle this problem and multiple related challenges (little to no labeled data suitable for training and evaluation, highly technical domain language that makes it impossible to use third-party data labeling services, such as Mechanical Turk [3], named entities that are too complex for the existing NER (Named Entity Recognition) tools), we are exploring the capabilities of large language models (LLMs) such as InstructGPT-175B and Flan-T5-XXL in few-shot and zero-shot learning via fine-tuning and prompt engineering, respectively. These models have shown promising results [4,5] in various natural language processing tasks and have the potential to extract relevant information from ATels and GCN circulars accurately. We have developed several end-to-end methods that rely on large language models and require less annotated data since they only need the annotation of entities to be extracted instead of every word in a text.

To make the results of our research publicly accessible, we are also developing a web application and publicly available server API that would provide all the data extracted with LLMs in a convenient way, along with the source code of the application. The beta version of the API is available at [6], and the web application will become available at the same address in the summer of 2023.

*1.1. Named Entity Recognition and Summarization*

1.1.1. Overview

Named Entity Recognition (NER) is an active research area that aims to identify and classify named entities in a text, such as people, organizations, locations, dates, etc. There are several approaches to NER, including rule-based [7], dictionary-based, and machine learning-based methods [8,9].

Flat NER refers to the task of identifying and classifying named entities in a given text without considering the relationships or hierarchies among the entities. This is in contrast to nested NER, also known as structured NER, which involves identifying and extracting hierarchically structured information from the text, such as relations between entities or their attributes. Nested NER is typically more challenging than flat NER, as it requires not only identifying the named entities in the text but also extracting the relationships and attributes between them.

Text summarization is the task of generating a shorter version of a document that preserves the important information and key points while reducing the length. There are several approaches to text summarization, including extractive and abstractive methods. Extractive summarization involves selecting a subset of the most important sentences or phrases from the original document to create the summary. These approaches are relatively simple to implement and can achieve good results, but they cannot paraphrase or rephrase the text. Abstractive summarization involves generating a summary that is a shorter version of the original document but may use different words and phrases to convey the same information. These approaches require the ability to understand the meaning and context of the text. They are also more prone to errors but can produce more fluent and coherent summaries that better capture the structure and key points of the document.

1.1.2. Existing Approaches

Several approaches have been proposed for flat NER. Some of the early techniques, such as rule-based [7] and dictionary-based systems, rely on a set of hand-crafted rules to identify and classify named entities. These systems typically use a combination of dictionary lookup, string matching, and heuristics to identify named entities in the text. While these systems can be effective, they are limited in their ability to handle novel or ambiguous named entities and require significant manual effort to develop and maintain.

Another approach is the use of machine learning-based methods. These methods typically involve the use of statistical models, such as hidden Markov models (e.g., Ref. [10]), maximum entropy models [11], and conditional random fields (e.g., Ref. [12]), to learn the characteristics of named entities and classify them based on those features.

Despite their effectiveness, these methods can be sensitive to the quality and quantity of the training data and can be difficult to adapt to new tasks or transfer to different domains. In recent years, progress has been made in improving NER using deep learning techniques, such as transformer-based models, convolutional neural networks, and recurrent neural networks, which can automatically learn hierarchical representations of the input text and make more accurate predictions. These models have achieved state-of-the-art results on many NER benchmarks and have the advantage of being able to handle long-range dependencies and contextual information in the input.

One of the recent works in the field of astronomical name entity recognition introduced TDAC, a Time-Domain Astrophysics Corpus for NLP [13]. Aside from collecting the first annotated and publicly available dataset of such kind (75 fully annotated time-domain astrophysical reports), the work also presents a series of experiments with fine-tuning of

LLMs based on Bidirectional Encoder Representations from Transformers (BERT) architecture [14] that have shown promising results. Another important takeaway from this work is the time-consuming nature of full-text annotation: authors highlight that it takes about 4.5 min for the PhD student to annotate a single document. Thus, we find it very important to explore end-to-end deep learning approaches that only require a single text label that is much easier to prepare compared to an annotation.

### 1.1.3. Challenges

There are several challenges when extracting named entities from astronomical texts. One is the use of technical terms, which can be unfamiliar to a language model and may not be included in its training data [15]. The peculiarity of a task in a specific domain language means the absence of existing labeled datasets suitable for training and evaluation, as well as the difficulty of outsourcing data for labeling. Additionally, there may be ambiguity in the text, such as when words refer to an event that was not observed in the process described in the astronomical telegram merely mentioned, making it challenging to output only relevant entities. One more difficulty connected to extracting specific entities is that a text may contain a named entity while not clearly stating it, e.g., we can infer that the event type in the text is an X-ray burst due to the telescope that was used in observation. Named entities can also have complex multi-word structures, where omitting one word can lead to the loss of information, e.g., low mass X-ray binary (LMXB) compared to simply binary.

### 1.2. *Large Language Models*
### 1.2.1. Overview

Large language models (LLMs) are advanced natural language processing tools that use machine learning to gain knowledge from massive datasets of human language texts. They are trained on vast amounts of text data and are able to perform a variety of tasks, such as language translation, text generation, and information extraction. LLMs have tens to hundreds of billions of trainable parameters.

In addition to their ability to learn from large amounts of data, LLMs also have the ability to perform few-shot and zero-shot learning. Few-shot learning involves training a model on a small amount of labeled data and then fine-tuning it for a specific task. Zero-shot learning goes a step further and involves a model's ability to recognize and classify new objects or categories without any training examples for those specific classes.

Both few-shot and zero-shot learning are helpful when there is a limited amount of labeled data available for a specific task, as is often the case in natural language processing. By leveraging the powerful capabilities of LLMs and techniques such as few-shot and zero-shot learning, it is possible to achieve accurate results even with limited data. LLMs are often built using the Transformer architecture, which is a type of neural network that uses self-attention to process sequential data. Transformer networks are highly efficient and have shown excellent results in a variety of natural language processing tasks. Self-attention allows the model to weigh the importance of different words in a given input and focus on the most relevant ones. This is achieved through the use of attention mechanisms, which assign a weight to each word based on its relevance to the task at hand.

### 1.2.2. Few-Shot Learning via Fine-Tuning

One of the most important properties of large language models is their ability to perform well in few-shot learning tasks. Few-shot learning refers to the ability of a model to perform well on a new task with only a small amount of labeled data. This is particularly useful in situations where labeled data is scarce or difficult to obtain, such as in the field of natural language astrophysical reports, where data collection can be challenging.

Large language models, such as InstructGPT-3, have been shown to have exceptional few-shot capabilities due to their pre-training on large amounts of diverse data. InstructGPT-3, for example, is pre-trained on a massive amount of text data from the internet, which allows it to understand the nuances and complexities of natural language.

This pre-training allows InstructGPT-3 to generalize to new tasks with minimal fine-tuning, which is the process of adapting a pre-trained model to a new task using a small amount of labeled data.

The few-shot learning capabilities of GPT-3 have been demonstrated in a number of studies. For example, in a study by Brown et al. (2020) [16], GPT-3 was fine-tuned on a small amount of labeled data to perform named entity recognition on scientific papers and was able to achieve comparable performance to models trained on much larger amounts of labeled data.

Another study by Raffel et al. (2019) [4] found that large text-to-text neural networks, such as T-5, are similarly able to perform well on a wide range of natural language understanding tasks with minimal fine-tuning. Unlike GPT-3, the T-5 was pre-trained on a diverse set of text-to-text tasks such as summarization, translation, and question-answering.

### 1.2.3. Zero-Shot Learning via Prompt Engineering

Another important property of large language models is their ability to perform zero-shot learning through prompt engineering [17]. Zero-shot learning, also known as "zero-shot transfer" or "meta-learning", is the ability of a model to perform a task without any fine-tuning (i.e., no gradient updates are performed) specific to that task. This is achieved through the use of carefully designed prompts that guide the model to perform the desired task.

For example, in the case of NER, a prompt could be designed to guide the model to identify specific entities such as "astronomical objects' names" or "astronomical event types" in astrophysical reports. Similarly, in the case of classification, a prompt could be designed to guide the model to classify reports based on their content.

We should note that there is some ambiguity about the term "zero-shot"—in some literature, it means "learning from zero examples", i.e., the ability of a model to perform a task without any labeled data specific to that task. However, that is not the case with prompt engineering, as it involves providing inference-time task-specific information to the model. Our paper explores zero-shot learning only in the sense that no updates of model weights are performed.

### 1.3. Decoding Strategies

Significant progress in natural language processing has not only led to advancements in Transformer architecture but also in decoding techniques. Decoding algorithms, such as the greedy search, select the highest probability word at each time step as the subsequent output token until reaching a maximum length or generating an end-of-sequence token. However, this approach fails to consider the entire sequence context, potentially resulting in suboptimal outputs. A more sophisticated decoding algorithm, beam search, addresses this limitation by considering a set of probable output sequences instead of choosing the highest probability word at every step. This process involves maintaining a fixed-size candidate set called the beam. The model generates a set of potential candidates at each time step, scores them based on their probability, and selects the top-K candidates to include in the beam. However, beam search also tends to produce repetitive or generic outputs and can be computationally intensive, particularly for longer sequences. Consequently, its application may be limited in real-time or resource-constrained scenarios. This issue has been extensively discussed in the language generation research of Vijayakumar et al. (2016) [18] and Shao et al. (2017) [19].

In addition to deterministic approaches, stochastic methods can be explored to prevent the output from being too generic. This can be achieved by using the probabilities generated by the softmax function to determine the next token. Random sampling methods are useful for generating more diverse and creative outputs while maintaining a balance between randomness and coherence in the generated text.

Temperature sampling [20] is one such method that adjusts the level of randomness in the generation of text by modifying the temperature hyperparameter of the probability

distribution. Specifically, dividing the logits by the temperature value decreases the probability of unlikely tokens while increasing the probability of likely tokens. The resulting values are then passed through the softmax function to obtain the probability distribution across the vocabulary. Lower temperature values encourage more deterministic behavior, whereas higher values promote greater diversity.

Top-K sampling [21] constrains the number of candidate words to the K most probable tokens, which ensures that less probable words are excluded from consideration. However, this method has limitations because it does not account for the shape of the probability distribution and filters the same number of words regardless of the distribution's shape. This constraint can result in nonsensical output for sharp distributions or limit the model's creativity for flat distributions.

The top-p sampling method [22] was introduced to enhance the top-k method by selecting the top tokens with a cumulative probability greater than a certain threshold (p), thus avoiding the generation of low-probability tokens. Lower values of p yield more focused text, while higher values promote exploration and flexibility.

While random sampling techniques can result in less cohesive text and potentially inaccurate responses, they do provide diversity in the generated content. By sampling a model n times, a range of answers may be obtained, including some correct ones.

To assess the accuracy of these responses, evaluation metrics such as top-1 and top-n accuracy can be utilized. Top-1 accuracy refers to the percentage of times the model's top-ranked output is correct, while top-n accuracy measures the proportion of correct answers within the model's top-n predictions. It has been noted that top-n accuracy is generally higher than top-1 accuracy. This finding suggests that refining the ranking process of multiple model outputs can enhance top-1 accuracy. Nevertheless, challenges remain, such as developing reliable confidence metrics for ranking model outputs.

## 2. Processing Astronomical Texts with InstructGPT

### 2.1. Overview

GPT-3 (Generative Pre-trained Transformer) [16] is a state-of-the-art language model developed by OpenAI. It is a Transformer-based model with 175 billion parameters that was trained on a dataset of over 570 GB of text data. GPT-3 is capable of generating human-like text and can perform a wide range of language tasks, such as language translation, summarization, and question answering, with high accuracy.

InstructGPT is a version of the GPT-3 that was fine-tuned with human feedback by OpenAI [5]. The InstructGPT is superior to GPT-3 in terms of following instructions in English due to the difference in language modeling objectives between these two models. Many LLMs are trained to predict the next token in a given text, while InstructGPT is designed to act in accordance with the user's objective. To achieve this result, OpenAI uses a three-step process. Firstly, they fine-tune a GPT-3 model using a dataset of prompts and human-written outputs. Next, a reward model is trained using human-ranked model sampled outputs. Finally, this reward model is used to fine-tune a supervised baseline using the PPO (Proximal Policy Optimization) algorithm. This approach results in InstructGPT being significantly preferred to GPT-3 when considering how well the model follows the user's instructions.

GPT-3 and, consequently, InstructGPT models have shown an understanding of astronomical terms required for our task during the initial experiments. This fact, along with these models achieving state-of-the-art results on a wide range of natural language processing (NLP) tasks, led us to choose them for this research.

Text-to-Text Transfer Transformer (T5) [4] is an open-source neural network architecture developed by Google Research for a wide range of natural language processing tasks. Unlike GPT-3, which uses a language modeling approach, T5 uses a text-to-text transfer learning framework, where the model is firstly pre-trained on the unsupervised text denoising objective, then fine-tuned to generate text that is a correct answer to a given text input. The task-specific information is encoded in the prompt, allowing the model to

adapt to different tasks without changing its architecture. The model architecture consists of a stack of transformer layers, similar to GPT-3, with an encoder-decoder structure where the input text is encoded into a latent representation and then decoded into output text.

Flan-T5 is a version of the T5 neural network that was additionally instruction-finetuned on 1836 tasks that were phrased as instructions and often included chain-of-thought reasoning annotations [23]. Aside from the increase in performance of task-based benchmarks, instruction finetuning on chain-of-thought data makes the model much more capable of zero-shot reasoning. In this paper, we exclusively examine Flan-T5-XXL, the biggest publicly available version of Flan-T5 with 11B parameters.

### 2.2. Zero-Shot InstructGPT 175B and Flan-T5-XXL 11B

Prompt engineering is a naive approach to extracting named entities with LLMs that is the easiest to start with. We construct a prompt for each category of extracted information and then iteratively improve it, using labeled data for evaluation. This approach is the most straightforward and most convenient to start the research with because it does not require extensive training data or complex architectures. It can be performed with a pre-trained model and a small amount of labeled data (used solely for the evaluation of the model), which makes it easy to implement. However, it is worth noting that prompt engineering can also have its limitations. For instance, the quality of the generated text is highly dependent on the quality of the prompts, and if the prompts are not well-written or do not provide enough context, the generated text may not be coherent or relevant.

To refine the model outputs, we tried several different prompting approaches. The resulting prompts can be divided into categories depending on the idea behind them, and the method that was used for creating them.

The first prompt type is a simple task description, in which we ask the model to extract a certain entity from a text (example shown on Figure 2). Due to GPT-3's extensive range of functions that do not require runtime learning, we anticipate that the zero-shot prompts will yield satisfactory outcomes [24]. As the completions tended to contain additional information, we specified at the end of prompts what to include in the answer. While this type of prompt had difficulties discerning between different entities and dealing with absent entities, it has shown good performance.

```
Extract the observed astronomical object from the text:  {Dataset
Message}
We don't need the name of the object, only its type, so output only one
thing — the observed astronomical object:
```

**Figure 2.** An example of description prompt for the type of the object or physical phenomena.

The second type of prompt is based on using examples in prompts, e.g., mentioning possible event types or astronomical object names. It was utilized to show the model the type of entities we wanted to see in the output - to help it differentiate between different entities. However, in case the required entity is absent in the astronomical observation, the model is prone to use one of the entities mentioned in the prompt, leading to the wrong answer. That led to a third type of prompt—"few-shot" prompts. These prompts contained the description of the task and several short astronomical texts with desired answers (example shown on Figure 3). This way, we can give a general description of the desired entity through examples as well as mention some borderline cases, such as the absence of the required entity.

```
Task:  Find the observed event type in the text.
T: The Large Area Telescope (LAT), one of two instruments on
the Fermi Gamma−ray Space Telescope, has observed increasing
 gamma−ray emission  from a source positionally consistent with the
very−high energy peaked  BL Lac object   1ES 1215+303 .
A: gamma−ray emission
T: Subjects:  Infra−Red, Supernovae, Transient.  Description:  The
Wide−field Infrared Survey Explorer (WISE; Wright et al.  2010)
obtained flux density measurements of five of the six currently known
members of the class of objects known as  Luminous Red Novae .
A: None
T: 10  SGR−like bursts  were detected by Konus−Wind and
Helicon−Coronas−F. The triangulation of several these bursts
indicated that their origin is  SGR 1806−20 .
A: SGR−like bursts
T: MASTER−Tunka auto−detection system (Lipunov et al., ''MASTER
Global Robotic Net'', Advances in Astronomy, 2010, 349171) discovered
 optical transient  at (RA, Dec) = 22h 44m 05.86s +43d 45m 32.2s on
2015−11−16.51056 UT.
A: optical transient
```

**Figure 3.** An example of a "few−shot" prompt for an event type. T. and A. denote input text and expected answer correspondingly. Named entities are marked with color: ■ object name; ■ the type of the object or physical phenomena; ■ event type.

In our experiments, this kind of prompt has allowed us to correctly identify dataset messages that contain no required named entities. However, the overall performance has decreased; thus, we refined them by adding short explanations of why such answers were chosen (example shown in Figure 4). By providing explanations along with the examples, the model can learn to understand the underlying reasoning or intent behind the classification, rather than just memorizing the examples [25]. Existing work in prompt tuning suggests that in-context supplemental information can affect the model performance positively.

```
A: SGR−like bursts
Explanation:  Konus−Wind and Helicon−Coronas−F observed the SGR−like
bursts.  SGR 1806−20 is the source of SGR−like bursts, it is not an
event.  Therefore, the observed event is SGR−like bursts.
```

**Figure 4.** An example of explanation inserted in "few−shot" prompt for the event type. A. denotes the expected answer.

The fifth type of prompts was obtained by sampling them from the language model using the Automatic Prompt Engineer (APE) method. The aim was to reduce the human effort in creating the prompts and automate the process so that it would be possible to apply it to new kinds of entities in the future. The meta-prompts for generation and resampling were taken from previous work in this area [26]. We chose 10 as our sample size, as the instructions generated by the model differ little in semantic content, and we are able to find adequate prompts under this constraint. The resulting prompts contain only the description of the task and no examples, compared to the previous prompts (shown in Figure 5). Despite this, zero-shot performed better than few-shot.

```
Input a description of an astronomical object.  Output the object's
classification
```

**Figure 5.** An example of prompt obtained by APE method for the type of the object or physical phenomena.

Taking into account the improved performance of zero-shot prompts, we decided to tune this kind of prompt. To direct a GPT-3's inference in the truth-seeking pattern, we employ reasoning in our prompts. Compared to few-shot prompts with explanations, this time, we ask the model to come up with a motivation for such output by asking the model via the prompt "Let's solve this problem by splitting into steps.", as shown in Figure 6. The term "chain-of-thought" prompt has been used to describe this type of prompt [27].

```
Astronomical event is something we can detect with a telescope.  For
example, it can be gamma-burst or transient signals.  Astronomical
events are the results of some physical phenomena, such as supernova
explosions or neutron-star binary mergers.  Extract the observed
astronomical event from the text:  {Dataset message}.  Let's solve
this problem by splitting it into steps.
```

**Figure 6.** An example of a prompt for coming up with motivation for an answer.

To receive an answer in a closed form that is convenient for the task, we inject "Thus, the correct answer is" after generating model output and query the model again with the new prompt.

### 2.3. Few-Shot GPT-3 175B

As another approach to the extraction of named entities using LLM, we fine-tuned the OpenAI davinci model—GPT-3 with 175B parameters—using OpenAI API. We have tried several approaches to fine-tuning.

The first idea is to retrieve all required entities at once, meaning the output for a given text message from the ATel dataset would look as shown in Figure 7—and it would either contain the corresponding entity if it is present in the text message or a tag 'none'. This method has shown the lowest accuracy, so we decided from this point on to extract only one entity at a time.

```
[Event ID or object name]
[Event type]
[Physical phenomena or astronomical object]
```

**Figure 7.** An example of output with three entities.

Thus, in the second experiment, we fine-tuned the model on training examples consisting of a dataset input example and one type of its associated named entity. This approach is more computably expensive, as we have to fine-tune a model for each of the entities we want to obtain. To address this issue, we changed the training examples for the fine-tuned model to be able to extract all types of entities. For this, we used the prompts with the best performance created in the previous section. The input during the fine-tuning consists of a dataset text message embedded in one of the prompts. The output to this input is chosen according to the type of entity this prompt is designed to extract. This way, to obtain a certain kind of entity during inference, we would have to prepend the astronomical text with the corresponding prompt. The first two datasets are of size 202, while the third one contains 606 message-entity pairs since for the last dataset, we used three prompts (for each entity) to prepend all 202 texts to fine-tune for all entities.

### 2.4. Embedding Ranking for InstructGPT

Due to the probabilistic nature of the sampling process, it is expected of LLMs to not be able to generate an accurate answer at 100% of attempts. Following the evaluation procedure as defined in Ref. [23], we sample our LLMs five times with temperature $T = 0.7$, and then rank completions based on their perplexity. When ranking LM's sampled outputs, perplexity can be used as a way to evaluate the quality of the generated text. Because perplexity is a measure of how well the model can predict the next word in a sequence, a lower perplexity score for an output indicates that the output is more likely to contain

coherent and grammatically correct text. Perplexity is calculated as the exponential of the average negative log-likelihood of the model's predictions. In our case, we expect the answer with the lowest perplexity to be the correct one.

After manually examining each of the generated completion, we found out that quite often, LLMs do generate the correct answer, but they do not necessarily have the best perplexity as shown in Table 1. We define Δ accuracy as the difference between the highest possible accuracy (if we were always selecting the correct completion, if present) vs. the accuracy obtained by the perplexity ranking. To maximize the accuracy of our NER process, we developed an alternative method of embedding ranking.

**Table 1.** The influence of multiple sampling attempts on the LLM's accuracy is shown by the example of the `text-davinci-003` model on the type of object or physical phenomena. Notice that ranking answers by their perplexity does not always lead to the correct prediction (if such a prediction exists).

| Model | Description | Example | Few Shot | Explanation | APE | Steps |
|---|---|---|---|---|---|---|
| Top-5 accuracy | 0.737 | 0.405 | 0.539 | 0.648 | 0.514 | 0.643 |
| Perplexity-ranked top-1 accuracy | 0.648 | 0.346 | 0.341 | 0.504 | 0.316 | 0.361 |
| Δ Accuracy | 0.089 | 0.059 | 0.198 | 0.144 | 0.198 | 0.282 |

We train a feed-forward network (FFN) that receives the embeddings of sampled answers obtained from a feature extractor network and predicts the embedding of a correct answer (Figure 8). The FFN consists of two linear layers combined with dropout layers. The experiments have shown that the most optimal hyper-parameters for training the FFN on a dataset consisting of 151 messages are the following: 12 epochs, $5 \times 10^{-4}$ for the learning rate of Adam optimizer. During the evaluation, we calculate the L2 distance between answer embedding and predicted embedding. The answer with the lowest L2 is assumed to be correct. During the training phase, we also minimize the triple margin loss of the L2 distance between embeddings with regard to the model's weights. This approach achieves the best accuracy compared to those described previously.

It should be noted that although training an FFN model for ranking may be less convenient, its primary advantage lies in having a smaller number of parameters. This results in reduced processing time and computing resources required to generate predictions, making it particularly useful in production environments with limited resources.

Increasing the model size or training dataset may decrease the gap between top-1 and top-n accuracies, as demonstrated in Jared Kaplan et al. (2020) [28]. However, despite this, we use different methods for ranking model outputs to reduce this gap due to recent research by Hyung Won Chung et al. (2022) [23], which indicates that achieving an accuracy improvement of 38.8% → 58.5% would require a considerable increase in LLM size (more than eight times larger).

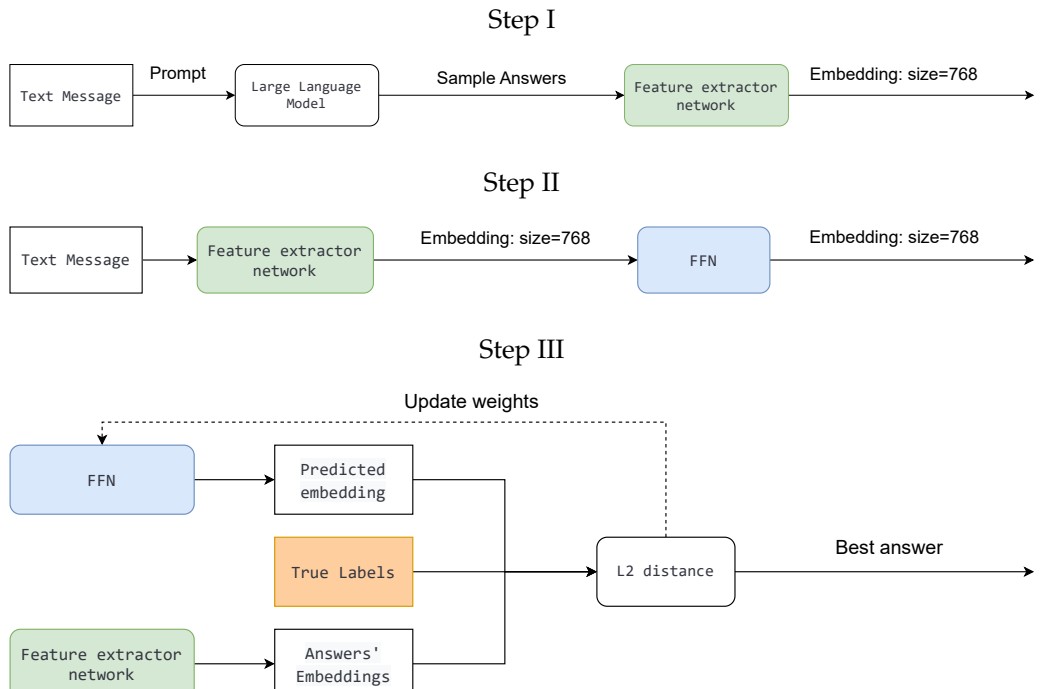

**Figure 8.** Steps of the embedding ranking pipeline (from up to bottom). Step I. Calculate the top three answers and their embeddings for a given text message and a prompt. Step II. Predict an embedding of a correct answer for a given text message. Step III. Calculate the L2 distance between answer embedding and predicted embedding. The answer with the lowest L2 is assumed to be correct.

## 3. Results

To understand how the model architecture and size affect the quality of output, given a certain prompt, we examine two davinci models available via the OpenAI API, and two FLAN-T5 models available via HuggingFace. To evaluate the quality of the prompts, we prepared a dataset of 202 messages with manually labeled ground truth answers for the following entities: event type; event ID or object name; physical phenomena, or astronomical object. For every text message from the dataset, we query a model five times and compare the predicted entities with the correct answers. Table 1 shows the inefficiency of perplexity-based answer ranking with respect to the accuracy of LLMs, using the example of `text-davinci-003` [29]. We define top-5 accuracy as the proportion of text messages in which at least one correct entity is present among the top five predictions. We define top-1 accuracy as the proportion of text messages in which the highest-ranked prediction (based on perplexity or embedding L2 distance) is the correct answer.

Our goal is to maximize top-5 accuracy, then additionally reduce the gap between top-5 and top-1 accuracies, which we label as Δ Accuracy. To achieve that, we compare two methods to rank sampled answers—perplexity ranking and embedding ranking using FFN. The comparison between best results using top-1 perplexity and embedding ranking is shown in Table 2. While being a common method to rank the LLM predictions (e.g., used by [23]), perplexity ranking produces suboptimal results compared to embeddings ranking.

As for the methods for comparing the model's outputs, the embedding ranking pipeline has shown the best performance compared to previous methods. As shown in Figure 9, we used MSELoss (minimizes squared L2 norm between embeddings of a prediction and correct answer) is represented by the solid line, ContrastiveLoss (minimizes squared L2 norm between embeddings of a prediction and correct answer, maximizes squared L2 norm between embeddings of a prediction and wrong answer) by the dashed line, and TripletMarginLoss (minimizes squared L2 norm between embeddings of a prediction and correct answer, maximizes squared L2 norm between embeddings of a prediction and wrong answer, plus adds a regularizing margin parameter). The highest accuracy is achieved by using TripletMarginLoss during training, with the hidden layers' dimension

of 512. The pipeline has shown itself to be suitable for practical tasks: it requires a low amount of labeled data (100–200 text messages) to learn a new entity category. However, it is computationally intensive.

**Table 2.** The comparison of models according to their top-1 accuracy. All models, except for the embedding ranking pipeline, have used perplexity ranking.

| Model | Event Type | Event ID or Object Name | Physical Phenomena or Astronomical Object |
|---|---|---|---|
| text-davinci-003 zero-shot | 0.495 | 0.801 | 0.648 |
| Flan-T5-XXL zero-shot | 0.544 | 0.861 | 0.514 |
| Fine-tuned GPT-3 | 0.677 | 0.806 | 0.548 |
| Embedding ranking pipeline | **0.941** | **0.980** | **0.823** |

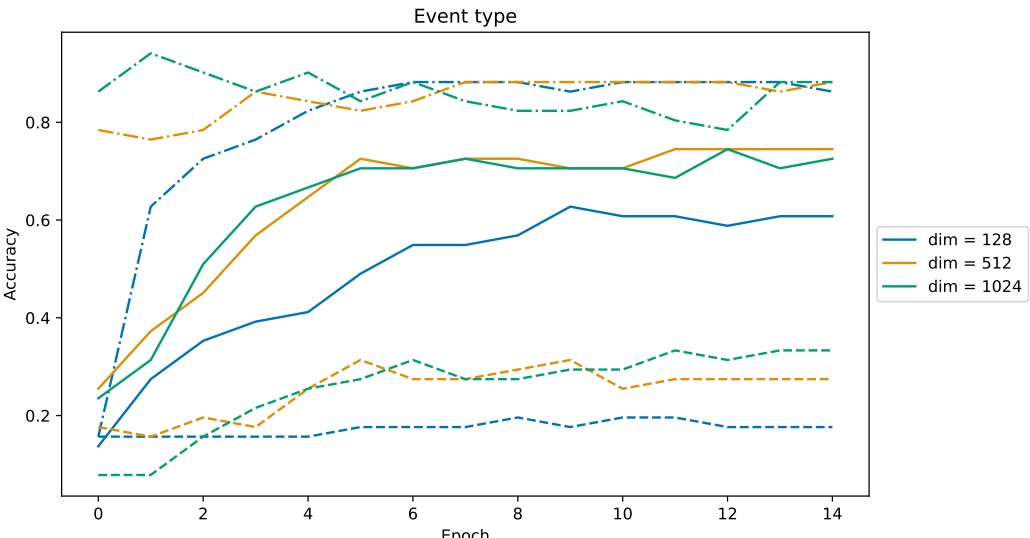

**Figure 9.** Comparison of FFN validation accuracy during training depending on a loss function and a hidden dimension size. Shown on the example of event type entity and fine-tuned `pearsonkyle/gpt2-exomachina` embeddings. The MSELoss is represented by the solid line, ContrastiveLoss by the dashed line, and TripletMarginLoss by the line with a dot.

To optimize the amount of computing power used for each prediction, as well as to improve the embedding quality, we tried to use different feature extractors for our embedding ranking pipeline. We considered the following models: GPT-2, GPT-2 trained on data from NASA's Astrophysical Data System (GPT-2 Exo-Machina), Exo-Machina that was fine-tuned on our dataset, GPT-NEO 125M, and InstructGPT. The fine-tuned GPT-2 Exo-Machina, as shown in Table 3, has the best combination of computing cost and accuracy. It was fine-tuned on the dataset of 202 text messages with three prompts per message (606 message-prompt pairs).

We also investigated the impact of fine-tuning LLMs, specifically InstructGPT, on a small dataset. The testing datasets consisted of 31 text messages (15% of human marked-up dataset). Table 4 shows that the models fine-tuned only on one type of named entities have shown the best performance in two types of entities. However, the result is worse compared to zero-shot learning. As for the model fine-tuned to output three entities at once, the generated outputs have shown that the model failed to capture the difference between event type and astronomical objects from the training dataset alone. The model trained on the instructions showed a slight difference in accuracy from the best result and can be instead used for computational efficiency. A comparative study of different prompts is also presented in Appendix A.

**Table 3.** Top-1 accuracy of the embedding ranking pipeline depending on the feature extractor network. Large Language Model that was used for sampling answers—`text-davincii-002`.

| Feature Extractor Network | Event Type | Event ID or Object Name | Physical Phenomena or Astronomical Object |
|---|---|---|---|
| text-embedding-ada-001 | 0.882 | 0.960 | 0.764 |
| gpt-2 | 0.745 | 0.901 | **0.823** |
| pearsonkyle/gpt2-exomachina | **0.941** | 0.921 | 0.784 |
| pearsonkyle/gpt2-exomachina fine-tuned | 0.882 | **0.980** | **0.823** |
| gpt-neo-125 | 0.823 | 0.921 | 0.745 |

**Table 4.** Fine-tuned GPT models top-1 accuracy on datasets of 31 text messages.

| Approach | Event Type | Event ID or Object Name | Physical Phenomena or Astronomical Object |
|---|---|---|---|
| Fine-tuning: 3 Entities in completion | 0.529 | 0.627 | 0.392 |
| Fine-tuning: 1 Entity in completion | **0.705** | **0.862** | 0.588 |
| Fine-tuning: Prompt in input, 1 entity in completion | 0.638 | 0.793 | 0.574 |
| Zero-shot learning * | 0.625 | 0.851 | **0.606** |

* Best prompt results on fine-tuning test datasets.

## 4. Outlook

Large language models have proven to be extremely useful for the task of analyzing astronomical reports that are written in natural language. Through this research, we have already achieved a >95% accuracy rate in the extraction of astronomical object names and event types from our labeled dataset of astronomical reports. We are continuing to work on extracting other entities from the text, such as physical quantities and descriptions of observations, in order to extract and structure the information contained in these reports fully. At the same time, we are also planning to prepare more labeled data to further increase the quality of our LLMs. All the predictions obtained with our best-performing method—embedding ranking pipeline—are publicly available via search API [6] and GitHub repository [30]. Using this API, our functionality can be integrated with multi-messenger brokers and platforms, e.g., with Astro-COLIBRI [31,32], to provide complementary information for human-written messages. In the coming months, we are also planning to release the first version of the web application for searching and cross-referencing ATels and GCN circulars using the data extracted with our LLMs. We are welcoming comments, feedback, and requests from the community, as well as third-party contributions of any kind.

**Author Contributions:** Conceptualization, V.S.; methodology, V.S. and A.C.; software, A.C.; validation, V.S. and A.C.; formal analysis, V.S.; investigation, A.C.; resources, V.S.; data curation, A.C.; writing—original draft preparation, V.S. and A.C.; writing—review and editing, V.S.; visualization, A.C.; supervision, V.S.; project administration, V.S. All authors have read and agreed to the published version of the manuscript.

**Funding:** This research received no funding.

**Data Availability Statement:** This research is based on the publicly available data from The Astronomer's Telegram (https://astronomerstelegram.org/ (accessed on 28 February 2023)) and NASA GCN circulars (https://gcn.nasa.gov/circulars (accessed on 28 February 2023)). We are gradually releasing the source code and prediction results in our publicly available GitHub repository: https://github.com/JetBrains/lm-astronomy (accessed on 28 February 2023).

**Acknowledgments:** We thank S. Golovachev and A. Ivanov for their help with setting up the R&D infrastructure. We express our sincere gratitude to D. Kostunin for providing insightful commentaries. We thank OpenAI for providing the early access to their embedding API.

**Conflicts of Interest:** The authors declare no conflict of interest. The funders had no role in the design of the study; in the collection, analyses, or interpretation of data; in the writing of the manuscript; or in the decision to publish the results.

## Appendix A. Comparative Study of Different Prompts

Tables A1–A3 show the performance of different prompts on mentioned models. As expected, larger models have demonstrated overall better performance. Predictably, on a simpler named entity, such as an astronomical object name or event ID, all models and prompts have shown noticeably higher accuracy compared to the more complex entities that require text comprehension. For the latter entities, such as event type and the type of astronomical object or physical phenomena, instructions containing only the task description have shown better performance. As we noticed while working with this type of instruction, there has to be a clearly stated output format, or else the model output would be contaminated with unnecessary information.

**Table A1.** Accuracy of predicting the event ID or object name entity for different models and prompts. Evaluated on a dataset of 202 text messages. The first value in a column is top-5 accuracy, and the second one is perplexity-ranked top-1 accuracy.

| Model | Description | Example | Few Shot | Explanation | APE | Chain-of-Thought |
|---|---|---|---|---|---|---|
| text-davinci-002 | 0.920 0.742 | 0.915 0.811 | 0.861 0.688 | 0.876 0.732 | 0.722 0.643 | 0.846 0.618 |
| text-davinci-003 | 0.801 0.722 | 0.826 0.747 | 0.821 0.742 | 0.831 0.801 | 0.708 0.698 | 0.772 0.683 |
| flan-t5-xl | 0.925 0.846 | 0.891 0.836 | 0.831 0.752 | 0.707 0.623 | 0.741 0.663 | 0.881 0.737 |
| flan-t5-xxl | 0.930 0.861 | 0.910 0.851 | 0.851 0.772 | 0.727 0.658 | 0.738 0.722 | 0.866 0.663 |

**Table A2.** Accuracy of predicting the physical phenomena or astronomical object for different models and prompts. Evaluated on a dataset of 202 text messages. The first value in a column is top-5 accuracy, and the second one is perplexity-ranked top-1 accuracy.

| Model | Description | Example | Few Shot | Explanation | APE | Chain-of-Thought |
|---|---|---|---|---|---|---|
| text-davinci-002 | 0.737 0.470 | 0.346 0.123 | 0.559 0.193 | 0.445 0.158 | 0.683 0.321 | 0.772 0.415 |
| text-davinci-003 | 0.737 0.648 | 0.405 0.346 | 0.539 0.341 | 0.648 0.504 | 0.514 0.316 | 0.643 0.361 |
| flan-t5-xl | 0.044 0.014 | 0.148 0.128 | 0.227 0.113 | 0.075 0.108 | 0.663 0.400 | 0.732 0.475 |
| flan-t5-xxl | 0.430 0.366 | 0.272 0.212 | 0.633 0.514 | 0.308 0.410 | 0.504 0.272 | 0.658 0.430 |

**Table A3.** Accuracy of predicting the event type for different models and prompts. Evaluated on a dataset of 202 text messages. The first value in a column is top-5 accuracy, and the second one is perplexity-ranked top-1 accuracy.

| Model | Description | Example | Few Shot | Explanation | APE | Chain-of-Thought |
|---|---|---|---|---|---|---|
| text-davinci-002 | 0.579 | 0.321 | 0.158 | 0.381 | 0.410 | 0.688 |
| | 0.381 | 0.099 | 0.054 | 0.198 | 0.158 | 0.366 |
| text-davinci-003 | 0.559 | 0.579 | 0.415 | 0.410 | 0.227 | 0.658 |
| | 0.495 | 0.460 | 0.262 | 0.267 | 0.099 | 0.301 |
| flan-t5-xl | 0.688 | 0.480 | 0.207 | 0.782 | 0.207 | 0.267 |
| | 0.524 | 0.356 | 0.103 | 0.603 | 0.059 | 0.227 |
| flan-t5-xxl | 0.663 | 0.232 | 0.222 | 0.801 | 0.603 | 0.415 |
| | 0.514 | 0.168 | 0.158 | 0.544 | 0.440 | 0.351 |

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
