# Peer review of "Language Models for Multimessenger Astronomy"

_galaxies, doi:10.3390/galaxies11030063_

Round 1

Reviewer 1 Report

The manuscript entitled "Language Models for Multimessenger Astronomy" describes and uses different machine learning techniques to obtain information from astronomical reports such as GCN circulars and ATels. This can help to automatize the association of phenomena with specific objects in the sky from different follow-up observations.

The text is clear and gives enough background for researches that are not familiar with the field of machine learning. The manuscript also specifies the main problem clearly, which is that astronomical reports are no machine-readable, and a way to solve this. 

I list my comments below and also attach an annotated PDF with them:

- Abstract: Add  a couple of words saying these two algorithms are machine learning algorithms for people that are not familiar with them.

- Line 22: Depending on the editor, the acronyms usually go in the parentheses

- Line 24: in an ATel...

- Line 70: Change "in the coming months" to "in the summer of 2023" (or fall depending on your deadlines)

- Line 104: Delete acronym (CRFs) since it's not used later in the text

-  Line 118: Define BERT

- Line 187: Use the acronym NER

- Line 315-319: Rewrite this paragraph. Mention first what Perplexity measures and then how it is calculated

- Line 322: "quite often". Is there a value to quantify this?

- Line 324: "Figure 2" points to Table 2. Should it point to Table 1?

- Line 349: What's top-5 and top-1? Is it the number of times the models were queried?

- Line 353-356: This paragraph appears twice in the text

-Line 385: You could mention (or mark in the tables) which models give more than 95% accuracy

- Figure 9: To make the plot easy to read you can select a line style for each type of criteria  (e.g. solid for MSELoss, dotted for Contrastive Loss and  dot-line for TripletMarginLoss).Then you can use 3 colors for each value of the dimension. The label can have only the dimensions and the caption can describe what each of the line styles are. 

Are all tables' accuracy calculated with respect to the 202 messages? (except for table 4). There was a mention of a data set with 606 messages but it wasn't clear to me if the results of that one were shown.

Author Response

Response to Reviewer 1 Comments

Point 1: - Abstract: Add  a couple of words saying these two algorithms are machine learning algorithms for people that are not familiar with them.

Response 1: Added, please see lines 7-8 of the new revision.

Point 2: - Line 22: Depending on the editor, the acronyms usually go in the parentheses

Response 2: Fixed.

Point 3: - Line 24: in an ATel...

Response 3: Fixed.

Point 4: - Line 70: Change "in the coming months" to "in the summer of 2023" (or fall depending on your deadlines)

Response 4: Fixed.

Point 5: - Line 104: Delete acronym (CRFs) since it's not used later in the text

Response 5: Fixed.

Point 6: - Line 118: Define BERT

Response 6: Fixed.

Point 7: - Line 187: Use the acronym NER

Response 7: Fixed.

Point 8: - Line 315-319: Rewrite this paragraph. Mention first what Perplexity measures and then how it is calculated

Response 8: Fixed.

Point 9: - Line 322: "quite often". Is there a value to quantify this?

Response 9: Added an explanation that it is measured in Table 1 (please see line 376).

Point 10: - Line 324: "Figure 2" points to Table 2. Should it point to Table 1?

Response 10: Fixed.

Point 11: - Line 349: What's top-5 and top-1? Is it the number of times the models were queried?

Response 11: Added an explanation, please see lines 410-413.

Point 12: - Line 353-356: This paragraph appears twice in the text

Response 12: Fixed.

Point 13: - Line 385: You could mention (or mark in the tables) which models give more than 95% accuracy

Response 13: Marked in the tables + mentioned on lines 455-456.

Point 14: - Line Figure 9: To make the plot easy to read you can select a line style for each type of criteria  (e.g. solid for MSELoss, dotted for Contrastive Loss and  dot-line for TripletMarginLoss).Then you can use 3 colors for each value of the dimension. The label can have only the dimensions and the caption can describe what each of the line styles are.

Response 14: Fixed.

Point 15: - Are all tables' accuracy calculated with respect to the 202 messages? (except for table 4). There was a mention of a data set with 606 messages but it wasn't clear to me if the results of that one were shown.

Response 15: Added clarification on how we obtain 606 message-entity pairs on lines 361-362, added information on how this dataset is used on lines 439-440.

Reviewer 2 Report

The authors examine the use of large language models (LLMs) for extracting information from astronomical reports, and investigate the zero-shot and few-shot learning capabilities of LLMs to improve the accuracy of predictions. The manuscript is well written. But some terms should be clarified, like the ones in the legend of Figure 9, MSELoss,  ContrastiveLoss and so on. And also the terms and methods in tables should be given more details. Are the numbers in the tables presenting the percetage?

Author Response

Response to Reviewer 2 Comments

Point 1: But some terms should be clarified, like the ones in the legend of Figure 9, MSELoss,  ContrastiveLoss and so on.

Response 1: We added a subsection explaining the sampling process (lines 200-247), as well as the explanation of differences between the loss functions used (lines 422-428).

Point 2: And also the terms and methods in tables should be given more details. Are the numbers in the tables presenting the percetage?

Response 2: We added an explanation of metrics shown on tables, please see lines 376, 410-413.

Reviewer 3 Report

The paper "Language Models for Multimessenger Astronomy" is inspiring and exciting. The method improves the development of data-driven applications for astrophysical text analysis using the IA Language Models. The text is well described and didactic, with interesting methodology and language models used and compared.

Minor points:

1. Missing references in some paragraphs. This can be improved across the text, please. For example:

a. "These models have shown promising results in various natural language processing tasks and have the potential to extract relevant information from ATels and GCN circulars accurately [REF...]."

b. Named Entity Recognition (NER) is an active research area that aims to identify and classify named entities in a text, such as people, organizations, locations, dates, etc. There are several approaches to NER, including rule-based, dictionary-based, and machine learning-based methods [REF...].

2. What worries me is basing results on a not open source code, as is the case with GPT-3. Can we say that the results are really consistent? This is a reflection. On the other hand, I see a promising future for astronomy using AI. The interesting results of this paper can observe this.

After improving the references, I indicate the paper for publication in Galaxies. May it be the first analysis of many to come.

Author Response

Response to Reviewer 3 Comments

Point 1: Missing references in some paragraphs

Response 1: We provided 12 additional references for better clarity in the new revision of the article.

Point 2: What worries me is basing results on a not open source code, as is the case with GPT-3. Can we say that the results are really consistent? This is a reflection. On the other hand, I see a promising future for astronomy using AI. The interesting results of this paper can observe this.

Response 2: The second family of models that we've used in our research (namely, Flan-T5) is open-source. This hasn't been mentioned in the paper, we updated the abstract and the section about T5 models with clarifications (please see lines 8 and 270 of the new revision).